# Exercise-Induced ADAR2 Protects against Nonalcoholic Fatty Liver Disease through miR-34a

**DOI:** 10.3390/nu15010121

**Published:** 2022-12-27

**Authors:** Zhijing Wang, Yaru Zhu, Lu Xia, Jing Li, Meiyi Song, Changqing Yang

**Affiliations:** 1Department of Gastroenterology and Hepatology, Tongji Hospital, School of Medicine, Tongji University, Shanghai 200092, China; 2Department of Urology, Tongji Hospital, School of Medicine, Tongji University, Shanghai 200092, China

**Keywords:** NAFLD, exercise, ADAR2, miR-34a

## Abstract

Nonalcoholic fatty liver disease (NAFLD) is a growing health problem that is closely associated with insulin resistance and hereditary susceptibility. Exercise is a beneficial approach to NAFLD. However, the relief mechanism of exercise training is still unknown. In this study, mice on a normal diet or a high-fat diet (HFD), combined with Nω-nitro-L-arginine methyl ester, hydrochloride (L-NAME) mice, were either kept sedentary or were subjected to a 12-week exercise running scheme. We found that exercise reduced liver steatosis in mice with diet-induced NAFLD. The hepatic adenosine deaminases acting on RNA 2 (ADAR2) were downregulated in NAFLD and were upregulated in the liver after 12-week exercise. Next, overexpression of ADAR2 inhibited and suppression promoted lipogenesis in HepG2 cells treated with oleic acid (OA), respectively. We found that ADAR2 could down-regulate mature miR-34a in hepatocytes. Functional reverse experiments further proved that miR-34a mimicry eliminated the suppression of ADAR2 overexpression in lipogenesis in vitro. Moreover, miR-34a inhibition and mimicry could also affect lipogenesis in hepatocytes. In conclusion, exercise-induced ADAR2 protects against lipogenesis during NAFLD by editing miR-34a. RNA editing mediated by ADAR2 may be a promising therapeutic candidate for NAFLD.

## 1. Introduction

Nonalcoholic fatty liver disease (NAFLD) threatens the health of human beings, has caused serious burdens on society, and has economic impacts. Recently, its concept has been updated to metabolic dysfunction-associated fatty liver disease (MAFLD) in order to directly reflect the characteristics of metabolic disorders [1]. In 2018, the estimation of the global prevalence of NAFLD was about 25% [2]. With the acceleration of lifestyle changes, the prevalence of NAFLD is continuing to increase in developing regions. The pathological process of NAFLD involves simple steatosis without liver injury, steatohepatitis, and end-stage liver disease. The excessive accumulation of triglycerides (TG) in the liver is thought to be an early manifestation of NAFLD [3]. As it is a disease related to heredity, environment, and metabolism, the pathogenesis of NAFLD is affected by a variety of factors, such as genetic, environmental, immune, and nutritional factors [4]. Importantly, effective methods of preventing and reversing NAFLD still need to be further explored.

Lifestyle modifications play a vital role in the treatment of NAFLD [5]. Exercise, as a method of lifestyle modification, is beneficial for patients with NAFLD [6,7]. Exercise can alleviate NAFLD by regulating lipid metabolism, lipophagy, insulin resistance, and inflammatory response [8,9,10,11]. Moreover, the composition and function of the gut microbiota are also closely related to the management of NAFLD by means of physical activity [12]. It has been reported that different forms of exercise, such as aerobic exercise, resistance exercise, or high-intensity intermittent exercise, are equally effective in reducing the intrahepatic triglyceride content in NAFLD patients [13]. Additionally, different exercise intensities, such as vigorous and moderate exercise, also show similar results in the elimination of lipid droplets of NAFLD patients [14]. However, the specific molecular mechanisms of the protective effect of exercise in NAFLD remain unknown.

A-to-I RNA editing is one of the most widely distributed types of modifications in the mammalian transcriptome [15]. The adenosine deaminases acting on RNA(ADAR) is the catalytic enzyme performing this modification. ADAR can deaminate adenosine (A) to inosine (I) by binding to the double-stranded RNA regions of protein-coding genes or noncoding sequences [16]. There are three types of ADARs in mammalian cells: ADAR1 (ADAR), ADAR2 (ADARB1), and ADAR3 (ADARB2). The phenomenon of RNA editing enables the generation of thousands of different RNAs from the same gene sequence, which greatly promotes the diversity and plasticity of the genetic information at the RNA level [17]. The dysregulation of ADAR is closely associated with many diseases, such as tumors, pulmonary fibrosis, and cardiovascular diseases [18,19,20,21]. A-to-I RNA editing is also a pivotal mechanism in the regulation of a proper phase response to light in the circadian clockwork [22,23]. It has been reported that ADAR1 functions as an oncogene, while ADAR2 has tumor-suppressive abilities in hepatocellular carcinoma. The differentially expressed ADARs in hepatocellular carcinoma have significant prognostic value and diagnostic capacity [24]. However, there are still no studies on the critical role of ADAR in NAFLD, and the underlying regulatory mechanisms also remain confusing. Therefore, we investigated the role of ADAR in exercise-induced protection against NAFLD.

MicroRNA (miRNA) is a type of non-coding single-stranded RNA that is encoded by endogenous genes that are about 19–25nt in length and that mainly regulate the expression of target genes by participating in post-transcriptional gene regulation [25]. MiRNAs, including miR-122, miR-29, and miR-21, have been well-established as crucial regulators in NAFLD [26,27]. In our previous study, we found that miR-212 and miR-149 were elevated in the livers of NAFLD mice fed an HFD diet, while exercise could reduce liver steatosis via the decline of miR-212 and miR-149 [28,29]. Moreover, the microarray assay results showed that miR-34a was also elevated in the livers of HFD-treated mice and that exercise could reduce lipogenesis through the downregulation of miR-34a [28]. Some primary miRNAs (pri-miRNAs) can be edited by ADAR, which may lead to the suppression or alteration of target molecules of mature miRNAs [30]. For example, miR-589–3p is edited by ADAR2 in normal brain cortices compared to de novo glioblastoma, which changes the target gene of miR-34a from the protocadherin 9 (PCDH9) to the metalloprotease Metalloproteinase 12 (ADAM12) [18]. MiR-379-5p, which has been edited by ADRA2, can inhibit tumor cell proliferation and promote apoptosis through the target gene CD97 [31]. In the heart, elevated ADAR2 negatively regulates mature miR-34a and protects against acute myocardial infarction and doxorubicin-induced cardiotoxicity after exercise [21]. Therefore, we speculate that ADAR2 can reduce lipid droplet accumulation by downregulating the miR-34a in NAFLD.

In the present study, we demonstrate that hepatic ADAR2 is downregulated in NAFLD mice, while exercise protects the liver from hepatic steatosis via the upregulation of ADAR2. Our study further shows that ADAR2 participates in the lipogenesis in HepG2 cells by targeting miR-34a, which is a crucial regulator of lipid metabolism. This research provides compelling evidence that increasing ADAR2 might be a potential strategy for managing NAFLD.

## 2. Materials and Methods

### 2.1. Animals and Treatments

Male C57BL/6 wild-type mice that were 8 weeks old were obtained from SLAC Laboratory Animal Company (Shanghai, China). The animals used in the study were housed on a 12 h light/dark cycle at 25 °C and were provided with free access to commercial rodent chow and tap water or to a high-fat diet (HFD) and Nω-nitro-L-arginine methyl ester, hydrochloride (L-NAME) for 12 weeks, a NOS inhibitor, which was shown to exacerbates liver injury in an obese rodent model of NAFLD [32]. The HFD was a rodent diet with 60 kcal% fat (D1249, Research Diet, New Brunswick, NJ, USA). L-NAME was dissolved in drinking water at a concentration of 0.5 g/L (N5751, Sigma-Aldrich, St. Louis, MO, USA). Mice were randomized into four groups: (i) control group, mice fed with standard chow and tap water; (ii) exercise groups, mice fed with standard chow and tap water and subjected to exercise; (iii) HFD and L-NAME diet group, mice fed with HFD and L-NAME; and (iv) HFD and L-NAME diet and exercise group. Each group contained 6 mice, and we used 24 mice in total in the research. The exercise plan was implemented as previously described [28,33]. The exercise mice were put on a treadmill that was specially designed for mice. The running training began at 5 m/min for 10 min with a speed and time increase of 2 m/min and 10 min each day. By the end of the experiment, the mice had been running at the speed of 15 m/min for 60 min for a period of 12 weeks. All training sessions were held at the same time each day to avoid effects on training performance. After the 12-week feed and exercise program, all animals were anesthetized and sacrificed. Liver tissues were isolated and snap-frozen for future analysis or were put into 4% paraformaldehyde buffer (PFA) immediately for histological study. All animal experiments were reviewed and approved by the Animal Ethics Commission of Shanghai Tongji Hospital, Tongji University School of Medicine (2021-DW-007).

### 2.2. Histological Analysis

Histological hematoxylin–eosin (H&E) staining and Oil Red O staining were used to elevate the lipid composition in the liver. Liver tissues from the mouse were cut into 4 μM sections for H&E staining after being fixed in 4% PFA and embedded in paraffin. Frozen 4 μM liver sections were washed with 60% isopropyl alcohol and then stained with Oli Red O solution for 8 min. Hematoxylin was used to stain the nucleus for 2 min. Finally, the sections were mounted using glycerol jelly as the mounting medium. Images were obtained using a microscope (Leica, Wetzlar, Germany).

### 2.3. Cell Culture and Treatment

Cells from human hepatocellular cancer cell line HepG2 were acquired from FuHeng Cell Center (Shanghai, China). High-glucose DMEM (Hyclone, Logan, UT, USA) containing 10% fetal bovine serum (Thermo Fisher, Waltham, MA, USA) and 1% penicillin/streptomycin was used as a growth medium for HepG2 cell cultures. The cell cultures were maintained under standard conditions (37 °C, humidified atmosphere, 5% CO_2_). Oleic acid (OA) was bought from Sigma. HepG2 cells were treated with a medium containing 250 μM OA for 24 h to induce lipogenesis in vitro.

### 2.4. Vector Construction

For the overexpression of human ADAR2, the coding region of human ADAR2 was obtained from the National Center for Biotechnology Information (NCBI). Gene fragments were generated by PCR amplification and cloned into FUGW. Finally, the fragments were verified using Sanger sequencing. The following PCR primers were used:

humanADAR2 Forward Primer:

5′-GGGACCGGTATGGATATAGAAGATGAAGAAAA-3′

humanADAR2 Reverse Primer:

5′-CCGGAATTCTCAGGGCGTGAGTGA-3′

Short interfering RNAs (siRNAs) were purchased from Hanbio Biotechnology (Shanghai, China).

### 2.5. Vector Transfection

HepG2 cells were precultured in serum-free media overnight, 8 h before transfection. MiR-34a inhibitor (100 nM), mimic (50 nM), and negative controls (RiboBio, Guangzhou China) as well as ADAR2 overexpression plasmid and ADAR2 siRNA were all transfected for 48 h using Lipofectamine 2000 (Invitrogen, Carlsbad, CA, USA) according to the manufacturer’s instructions.

### 2.6. Nile Red Staining

Nile Red staining was used to assess lipid droplet formation in cells. HepG2 cells were seeded in 24-well plates and treated with OA for 24 h (250 μM). After being washed with PBS, the cells were fixed with 4% PFA for 15 min at room temperature. Then, the cells were stained with Nile Red solution (0.1 mM) for 15 min. Cell nuclei were stained with DAPI (Peprotech, Rocky Hill, NJ, USA) for 15 min. The entire staining process was carried out away from light.

### 2.7. Quantification of TGs

Triglycerides (TGs) were measured according to the instructions of the assay kit (Nanjing Jiancheng Bioengineering Institute, China). TGs can be catalyzed to red quinones under the guidance of instruction. The color of the quinones is proportional to the level of TG. The protein concentrations were measured using the BCA protein assay kit (KeyGEN, China). The lipid content was calculated according to the lipid level/protein concentration.

### 2.8. Real-Time Quantitative PCR

Total RNA was extracted using Trizol (Invitrogen, Waltham, MA, USA) and then used for cDNA synthesis according to the instructions (TaKaRa, Kusatsu, Japan). cDNA was used as the template for qRT-PCR using SYBR-Green (TaKaRa, Japan). GAPDH was used as an internal control. The expression of miR-34a was detected by the Bulge-LoopTM miRNA qPCR Primer Set (RiboBio, Guangzhou, China) with SYBR-Green (TaKaRa, Japan) by qRT-PCR. U6 (RiboBio, Guangzhou, China) was used as the internal control for miR-34a. The relative expression levels of the genes were calculated using the 2−ΔΔCt method. The sequences of the primers used in the study are shown in Appendix A.

### 2.9. Western Blot Analysis

Proteins from the liver tissues were collected using RIPA buffer (Thermo Fisher Scientific, Waltham, MA, USA). The protein concentration was elevated using the BCA Protein Assay Kit. Total proteins were separated with 10% SDS-PAGE gels, transferred into PVDF membranes, and incubated with the primary antibodies at 4 ℃ overnight. The primary antibodies were as follows: ADAR2 (diluted 1:1000; A16748, ABclonal, Wuhan, China) and β-actin (diluted 1:100,000; AC026, ABclonal, Wuhan, China). After incubation with the secondary antibody, the proteins were visualized using a luminescent imaging system (BioRad, Hercules, CA, USA) and enhanced using the ECL Chemiluminescent Kit (Thermo Pierce, Waltham, MA, USA). The analysis of the protein bands was carried out using Image J software.

### 2.10. Statistical Analysis

The data in the paper are presented as means ± SDs. Either a t-test or two-way analysis of variance was performed on the samples to assess significant differences using GraphPad Prism version 7.0. A *p* value of < 0.05 was considered statistically different.

## 3. Results

### 3.1. Exercise Mitigates Liver Steatosis in Mice

To determine the protective effect of exercise on liver steatosis, mice were prescribed a high-fat diet and treadmill running for 12 weeks simultaneously (Figure 1A). We observed a significant increase in body weight in mice on HFD + L-NAME compared to the controls from the third week to the end of the experiment, while exercise training markedly reduced body weight (Figure 1B). Moreover, exercise also reduced the liver weight and the ratio of liver weight to tibia length in NAFLD mice (Appendix A). HE and Oil Red O staining indicated that the amount of lipid droplets in the liver tissues decreased in the NAFLD exercise group (Figure 1C,D). Fatty acid synthase (Fasn) and Sterol regulatory element binding protein-1c (Srebp1c) are regarded as markers of lipogenesis. Increased mRNA expression levels of these genes were detected in mice fed a high-fat diet, and those were reduced by exercise (Figure 1E). Collectively, all of the above data indicate that exercise can ameliorate liver steatosis in diet-induced NAFLD mice.

### 3.2. Hepatic ADAR2 Was Ameliorated by Exercise in the Group with Diet-Induced NAFLD Mice

ADAR2 is an important RNA-editing enzyme during the process of post-transcription. To identify the role of ADAR2 on lipid deposits, we examined the expression of ADAR2 in NAFLD mice after exercise. We observed that ADAR2 decreased in NAFLD and could also be amplified by running training (Figure 2A,B). Consistent results were obtained for mRNA and protein detection. Collectively, these data confirmed that ADAR2 expression was inhibited in NAFLD and could be induced by exercise, which was associated with hepatic lipid accumulation suggesting a role of ADAR2 in lipid metabolism.

### 3.3. ADAR2 Prevents Hepatocyte from Lipid Accumulation In Vitro

To investigate the effects of ADAR2 on hepatic lipogenesis, loss-of-function and gain-of-function analyses were performed on HepG2 cells. First, the efficiency of the overexpressing (OE) ADAR2 plasmid was confirmed. The ADAR2 level was significantly increased after transfection with the ADAR2 overexpression plasmid in HepG2 cells (Figure 3A). We used siRNA to induce the silence of ADAR2, and si-ADAR2-2 was chosen among three sequences due to its high knockdown efficiency (Figure 4A). The expression of ADAR2 mRNA was approximately two-fold lower compared to the control group. Nile Red staining indicated that ADAR2 overexpression was reduced (Figure 3B) and that ADAR2 knockdown (Figure 4B) accelerated the lipid droplet formation of HepG2 cells in vitro. Moreover, ADAR2 overexpression decreased the levels of TG (Figure 3C), while ADAR2 knockdown could elevate the levels of TG (Figure 4C). Meanwhile, PCR analysis showed that ADAR2 overexpression downregulated lipid formation (Figure 3D) and ADAR2 knockdown promoted lipogenesis (Figure 4D). These results provide evidence that ADAR2 might be a crucial regulator of lipogenesis during NAFLD.

### 3.4. miR-34a Is Regulated by ADAR2 In Vitro and In Vivo

The A-to-I changes in miRNA transcripts can influence processing, thereby affecting miRNA expression. MiR-34a was identified as a potential regulator in NAFLD [34,35]. To investigate whether miR-34a is regulated by ADAR2 in lipogenesis, the miR-34a expression levels were detected in ADAR2 overexpression and knockdown cells that had both been treated and untreated with OA. It showed that ADAR2 negatively regulated miR-34a expression in HepG2 cells. ADAR2 overexpression could decrease (Figure 5A) while ADAR2 knockdown could increase (Figure 5B) miR-34a expression in cells that had received and not received OA treatment. Meanwhile, miR-34a was also found to be increased in the NAFLD mice, while exercise reduced the expression of miR-34a (Figure 5C).

We further examined whether miR-34a mediated the ADAR2-induced lipid formation in HepG2 cells. First, we proved that miR-34a mimic significantly increased the expression level of miR-34a (Figure 6A). Examination with Nile Red staining showed that elevated miR-34a caused a notable increase in lipid content, regardless of the inhibition of ADAR2 overexpression in HepG2 cells (Figure 6B). Additionally, miR-34a mimicking significantly blunted the suppression of ADAR2 overexpression of TG levels in HepG2 cells (Figure 6C). We also observed that ADAR2 overexpression reduced mRNA levels of lipogenesis (Fasn and Srebp1c), while miR-34a mimicking could reverse this effect (Figure 6D). All of the results collectively indicated that miR-34a was induced by ADAR2 in lipogenesis.

### 3.5. ADAR2 Inhibits Hepatocyte Lipid Droplet Accumulation by Regulating miR-34a In Vitro

As miR-34a is negatively regulated by ADAR2 in hepatocytes, we further explored the role of miR-34a in hepatic lipogenesis. First, miR-34a was downregulated after transfection with miR-34a inhibitor (Figure 7A). As a result, miR-34a inhibitor decreased the lipid content (Figure 7B,C), whereas miR-34a mimicking raised the lipid content (Figure 8A,B) in HepG2 cells during OA treatment, as determined by Nile Red staining and the TG levels. Consistently, the protective effects of miR-34a inhibitor (Figure 7D) and the deterioration effect of miR-34a mimic (Figure 8C) in OA-induced lipid generation were also confirmed by the expression of Fasn and Srebp1c. Overall, our findings illustrated that miR-34a could induce lipid accumulation in HepG2 cells.

## 4. Discussion

NAFLD is a metabolic stress-induced liver injury that is closely related to insulin resistance and genetic predisposition and that can lead to liver injury, cirrhosis, and other death-related chronic liver diseases. Moreover, NAFLD is also closely associated with the high incidence of metabolic syndrome (MetS), type 2 diabetes mellitus (T2DM), arteriosclerotic cardiovascular disease, and colorectal tumors [36]. Due to the complicated molecular regulatory mechanism of NAFLD, there is still no specific treatment for NAFLD [37]. Lifestyle interventions such as exercise can promote the healthy state of NAFLD patients [38]. In our study, we demonstrate that hepatic steatosis in NAFLD mice induced by diet is significantly attenuated by aerobic exercise. Hepatic ADAR2 is downregulated in NAFLD mice, and it can be restored via aerobic exercise. The in vitro findings suggest that the elevation of ADAR2 reduces lipogenesis in HepG2 cells, while the decrease of ADAR2 drives lipogenesis induced by free fatty acids. Finally, we show that ADAR2 contributes to lipogenesis by targeting miR-34a during NAFLD development.

It is clear that lifestyle interventions are beneficial for NAFLD, especially for individuals with obesity, as recommended in clinical practice and supported by high-quality evidence [6]. Compared with drug treatment, exercise is a cost-effective intervention for both patients and susceptible populations with risk factors [6]. Some studies have shown that exercise cannot only recover serum levels of alanine aminotransferase (ALT) and aspartate aminotransferase (AST), but can also lower the hepatic lipid content, whether the exercise intervention is aerobic training, resistance training, or hybrid and acceleration training [38]. Aerobic exercise is an efficient means of alleviating NAFLD in mice and has been confirmed in animal models by several studies, and our previous study essentially showed the same result [28]. Different from high-intensity exercise training, aerobic exercise does not require prior adaptive training. [39]. Reducing intrahepatic lipid content may be one possible mechanism to alleviate NAFLD via exercise [38]. For example, exercise was able to reduce hepatic lipogenesis [40,41]. HFD combined with L-NAME was used to establish the NAFLD mice model, as previously described, representing a compound- and diet-induced NAFLD model [42]. L-NAME, the NOS inhibitor, can promote hepatic TG and cholesterol by regulating the metabolism of TG synthesis and cholesterol [32,43]. Typical manifestations of NAFLD have been found in mice, including body weight gain, liver weight gain, and improved liver steatosis. We trained the mice with moderate-intensity aerobic exercise. In our work, we found that exercise could attenuate the body weight and liver weight of mice. Moreover, exercise training significantly decreased intrahepatic lipid accumulation by reducing lipid synthesis in NAFLD mice.

The ADAR activity during A-to-I editing was considered due to its unique function on the diversity of RNA and protein to both coding and non-coding RNA [44]. It has been reported that ADAR1 plays a vital role in hepatic immune homeostasis, hepatocellular carcinoma, and adipogenesis [45,46,47]. ADAR2 is another member of the adenosine deaminase family. However, the influence of ADAR2 in lipogenesis, especially in NAFLD, is still unknown. Our results indicated that ADAR2 was downregulated in the livers of NAFLD mice, and that exercise resulted in an obvious reversal in the ADAR2 level, which is synchronous with the process of lipid accumulation and dissipation in the liver observed in the histology results. Then, we used a HepG2 cell model with OA treatment to further clarify the functional and molecular mechanisms of ADAR2 in lipid metabolism. Overexpression of ADAR2 in HepG2 cells could inhibit lipid generation, with a decrease in intracellular lipid droplets and the expression of lipogenesis-related genes. By contrast, the knockdown of ADAR2 promotes lipogenesis with elevated lipid accumulation in the adipocytes. The levels of TG in HepG2 cells were also decreased under the overexpression of ADAR2 compared to the OA treatment, while the inhibition of ADAR2 has the opposite effect. Together, we exposed the crucial role of ADAR2 in hepatic lipogenesis, which indicates that ADAR2 has the potential to be used as an exercise-related therapeutic target for NAFLD. Notably, the optimal exercise intensity and duration for interventions for metabolism-related diseases remains controversial. In our study, we explored the possible mechanism of ADAR2 in moderate-intensity exercise, but whether there are differences in the role of ADAR2 regarding different exercise prescriptions still needs elucidation by further research.

A-to-I editing is an abundant RNA modification that takes place in miRNA through ADARs. It plays a regulatory role in multiple processes, from miRNA generation to function. First, ADARs edit pri-miRNA or pre-miRNA to further influence the synthesis of mature miRNA because of their hairpin structures [48]. Moreover, A-to-I editing influences miRNA target recognition in mature miRNA [48]. Then, we identified the candidates of ADAR2 in the progression of NAFLD. First, we found a significant increase in the miR-34a in the liver of HFD-induced NAFLD mice in our previous research, and exercise reduced the expression of miR-34a [28]. At the same time, exercise-induced ADAR2 protected the heart from myocardial infarction and doxorubicin-induced cardiotoxicity via miR-34a in cardiomyocytes [21]. Three nucleotide sites of pri-miR-34a were changed after ADAR2 overexpression. In detail, U44-C and A61-G are two major editing sites of pri-miR-34a, and A72-G is a minor site in the antisense strand of pri-miR-34a. Moreover, ADAR2 overexpression changes the RT-PCR product of pri-miR-34a in cardiomyocytes, which further implies the RNA editing of pri-miR-34a [21]. To define the relationship between ADAR2 and miR-34a in NAFLD, we found that ADAR2 can negatively regulate the expression of miR-34a in vitro, regardless of OA treatment. Interestingly, in the mice models, we also showed that the miR-34a level in NAFLD mice was decreased. In contrast, a reversed miR-34a level was spotted in exercised NAFLD mice compared to in sedentary NAFLD mice, which further indirectly suggests that ADAR2 may change the expression levels of miR-34a inversely. Furthermore, miR-34a mimic was able to abolish the lipogenesis-reducing effect of ADAR2 overexpression in HepG2 cells. Overall, ADAR2 exerted its effect on lipogenesis by targeting miR-34a.

As a post-transcriptional regulator with extensive functions, miRNA has been confirmed to participate in the NAFLD process. Elevated levels of miR-34a were reported in both NAFLD and nonalcoholic steatohepatitis (NASH) mice in previous studies [49,50]. miR-34a can regulate liver steatosis by inhibiting very low-density lipoprotein secretion and by promoting hepatic oxidative stress and inflammation through different molecular structures such as hepatocyte nuclear factor 4, alpha (HNF4α), sirtuin1 (Sirt1), and cyclin-dependent kinase 6 (CDK6) [34,35,51]. Hepatic miR-34a expression was also increased in the serum of NAFLD and NASH patients [52]. In our work, we found that miR-34a played a crucial role in the development of NAFLD under the regulation of ADAR2. miR-34a inhibitor was capable of inhibiting lipogenesis with the decreased triglyceride accumulation in HepG2 cells, while miR-34a mimic was able to promote lipogenesis via the enhanced triglyceride accumulation in HepG2 cells in the presence of OA treatment.

To completely prove that ADAR2 contributes to the protective effect of NAFLD via regulating miR-34a as key molecules generated by exercise, further study is required to carry out in vivo research in mice and is required to evaluate the effect of ADAR2 as a therapeutic method for NAFLD. The expression of ADAR2 and its target mRNA/miRNA in human samples also requires further clarification. This could advance our understanding of ADAR2 in NAFLD development.

In conclusion, exercise-induced ADAR2 growth attenuates NAFLD development by decreasing the level of miR-34a. We provide experimental evidence that ADAR2 may present a potential therapeutic target of NAFLD.

## Figures and Tables

**Figure 1 nutrients-15-00121-f001:**
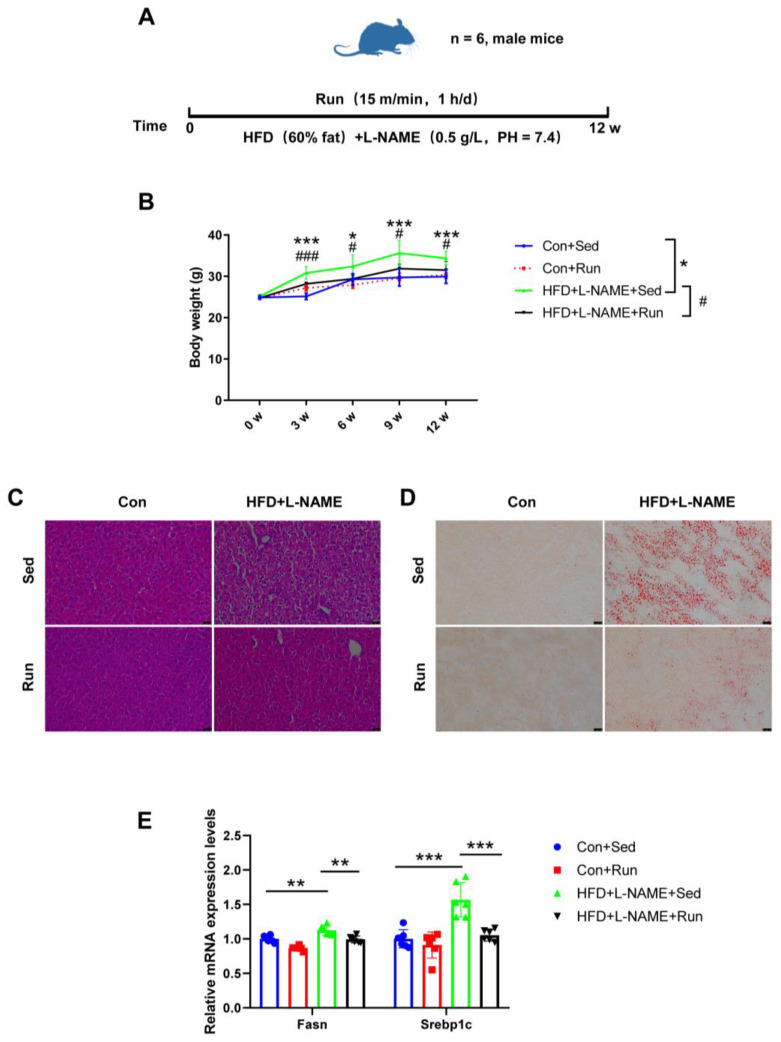
Exercise protects against liver steatosis in NAFLD mice. (**A**) The exercise schedule of NAFLD mice. (**B**) The body weight of mice. (**C**) The liver steatosis of mice determined by H&E staining. (**D**) The liver lipid content of mice detected by Oil Red O staining. (**E**) The mRNA levels of the genes related to liver lipogenesis (Fasn and Srebp1c) relative to GAPDH in the livers of mice. * *p* < 0.05, ** *p* < 0.01, *** *p* < 0.001; # *p* < 0.05, ### *p* < 0.001; *n* = 6. Sed, sedentary; HFD, high-fat diet; L-NAME, Nω-nitro-L-arginine methyl ester, hydrochloride; Fasn, fatty acid synthase; Srebp1c, sterol regulatory element binding protein-1c. Scale bar = 25 μm.

**Figure 2 nutrients-15-00121-f002:**
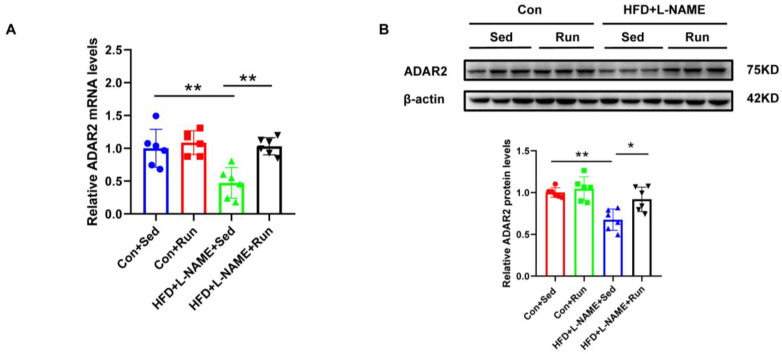
Hepatic ADAR2 is ameliorated by exercise in the group with diet-induced NAFLD mice. (**A**) The mRNA levels of ADAR2 relative to GAPDH in the mouse livers were analyzed by qRT-PCR. (**B**) The protein levels of ADAR2 in the mouse livers were detected using Western blotting. * *p* < 0.05, ** *p* < 0.01; *n* = 6. Sed, sedentary; HFD, high-fat diet; L-NAME, Nω-nitro-L-arginine methyl ester, hydrochloride; ADAR2, adenosine deaminases acting on RNA 2. ● Con+Sed; ■ Con+Run; ▲ HFD+L-NAME+Sed; ▼ HFD+L-NAME+Run.

**Figure 3 nutrients-15-00121-f003:**
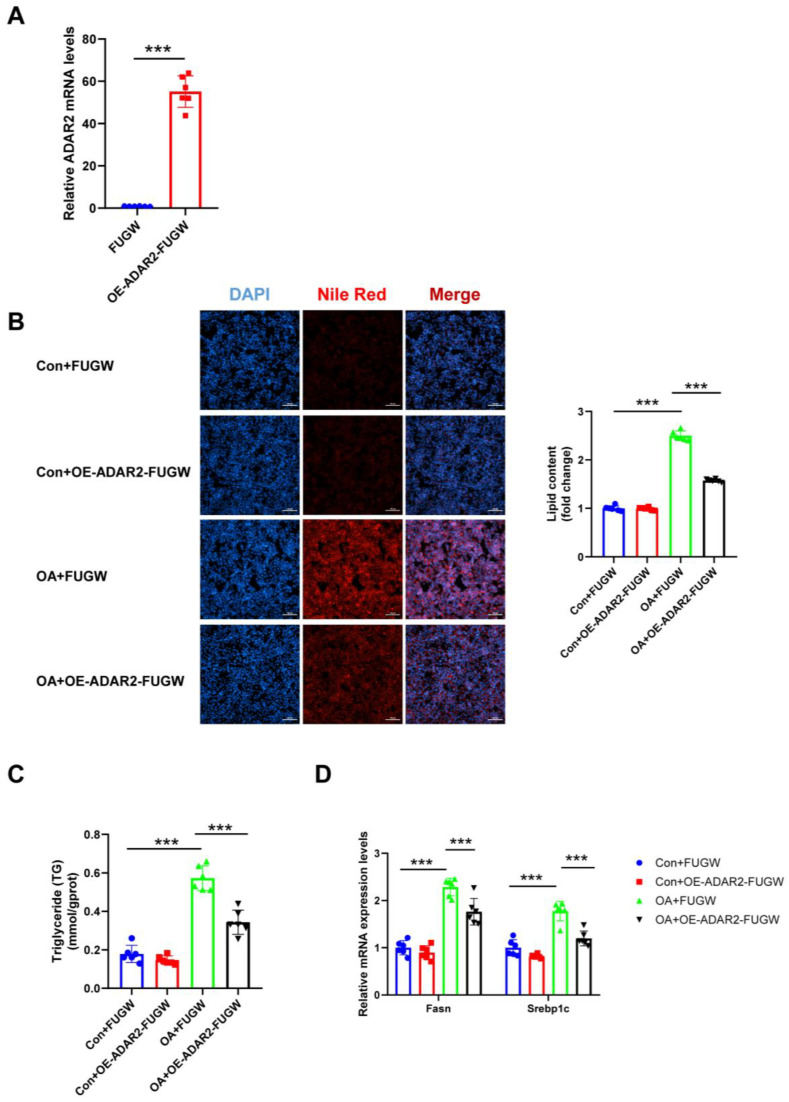
ADAR2 overexpression suppresses lipogenesis in hepatocytes. (**A**) The mRNA levels of ADAR2 relative to GAPDH were analyzed by qRT-PCR after transfection with the ADAR2 overexpression plasmid in HepG2 cells. (**B**) The lipid content of HepG2 cells was determined by Nile Red staining. (**C**) The TG levels of HepG2 cells. (**D**) The mRNA levels of lipogenesis-related genes (Fasn and Srebp1c) relative to GAPDH were analyzed by qRT-PCR. *** *p* < 0.001; *n* = 6. OE, overexpression; OA, oleic acid; TG, triglyceride; ADAR2, adenosine deaminases acting on RNA 2; Fasn, fatty acid synthase; Srebp1c, sterol regulatory element binding protein-1c. Scale bar = 100 μm.

**Figure 4 nutrients-15-00121-f004:**
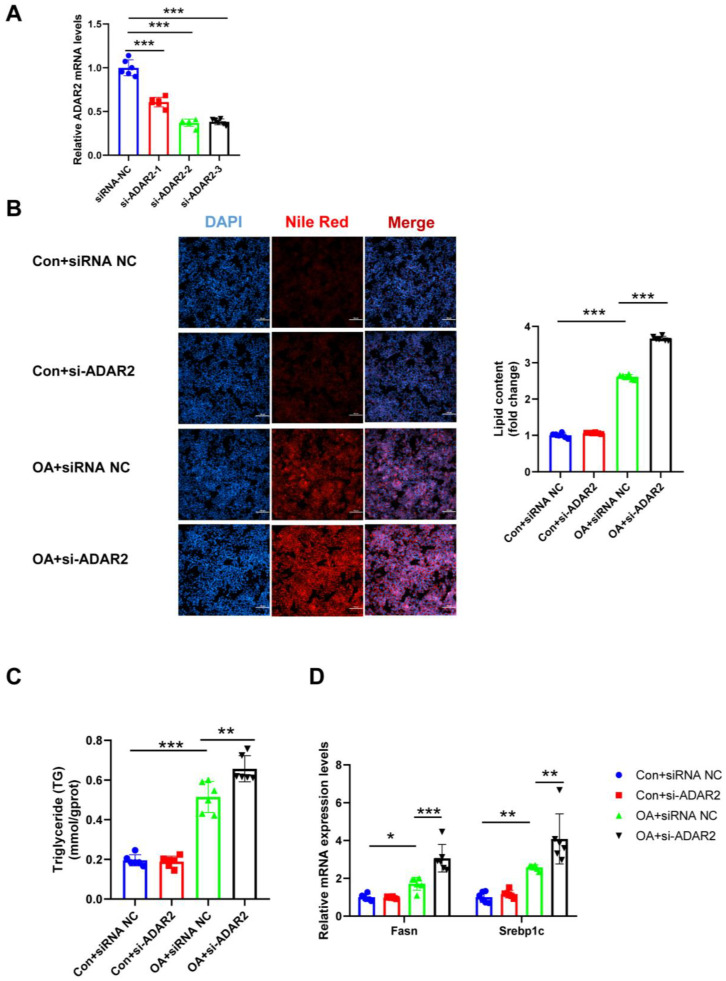
ADAR2 knockdown accelerates lipogenesis in hepatocytes. (**A**) The mRNA levels of ADAR2 relative to GAPDH were analyzed by qRT-PCR after transfection with ADAR2-siRNA in HepG2 cells. (**B**) The lipid content of HepG2 cells was determined by Nile Red staining. (**C**) The TG levels of HepG2 cells. (**D**) The mRNA levels of lipogenesis-related genes (Fasn and Srebp1c) relative to GAPDH were analyzed by qRT-PCR. * *p* < 0.05, ** *p* < 0.01, *** *p* < 0.001; *n* = 6. OA, oleic acid; TG, triglyceride; ADAR2, adenosine deaminases acting on RNA 2; Fasn, fatty acid synthase; Srebp1c, sterol regulatory element binding protein-1c. Scale bar = 100 μm.

**Figure 5 nutrients-15-00121-f005:**
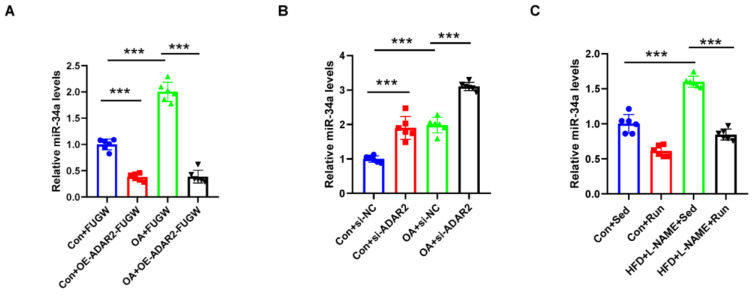
miR-34a is negatively regulated by ADAR2. (**A**) The miR-34a levels relative to U6 were analyzed by qRT-PCR after transfection with ADAR2 overexpression plasmid/vector in HepG2 cells. ● Con+FUGW; ■ Con+OE-ADAR2-FUGW; ▲ OA+FUGW; ▼ OA+OE-ADAR2-FUGW. (**B**) The miR-34a levels relative to U6 were analyzed by qRT-PCR after transfecting with ADAR2-siRNA/vector in HepG2 cells. ● Con+si-NC; ■ Con+si-ADAR2; ▲ OA+si-NC; ▼ OA+si-ADAR2. (**C**) The miR-34a levels relative to U6 in the livers of mice with NAFLD. ● Con+Sed; ■ Con+Run; ▲ HFD+L-NAME+Sed; ▼ HFD+L-NAME+Run. *** *p* < 0.001; *n* = 6. OE, overexpression; OA, oleic acid; Sed, sedentary; HFD, high-fat diet; L-NAME, Nω-nitro-L-arginine methyl ester, hydrochloride; ADAR2, adenosine deaminases acting on RNA 2.

**Figure 6 nutrients-15-00121-f006:**
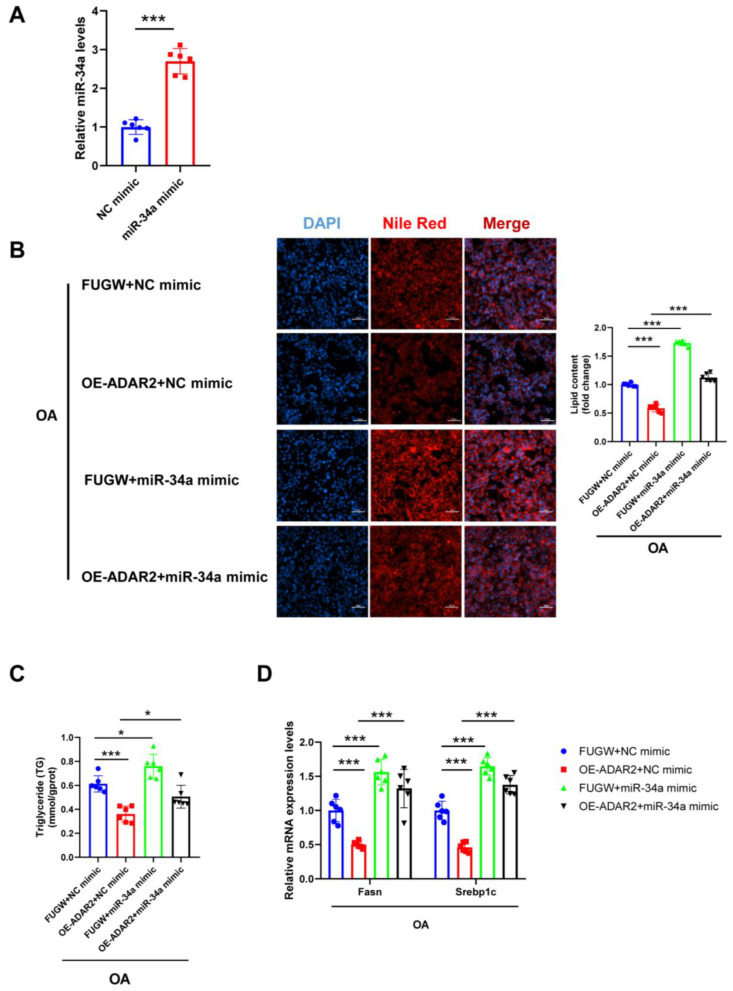
miR-34a blocks the inhibitory effect of ADAR2 on lipogenesis in hepatocytes. (**A**) The expression levels of miR-34a relative to U6 were analyzed by qRT-PCR after transfection with miR-34a mimic in HepG2 cells. (**B**) The lipid content of HepG2 cells was determined by Nile Red staining. (**C**) The TG levels of HepG2 cells. (**D**) The mRNA levels of lipogenesis-related genes (Fasn and Srebp1c) relative to GAPDH were analyzed by qRT-PCR. * *p* < 0.05, *** *p* < 0.001; *n* = 6. OE, overexpression; OA, oleic acid; TG, triglyceride; ADAR2, adenosine deaminases acting on RNA 2; Fasn, fatty acid synthase; Srebp1c, sterol regulatory element binding protein-1c. Scale bar = 100 μm.

**Figure 7 nutrients-15-00121-f007:**
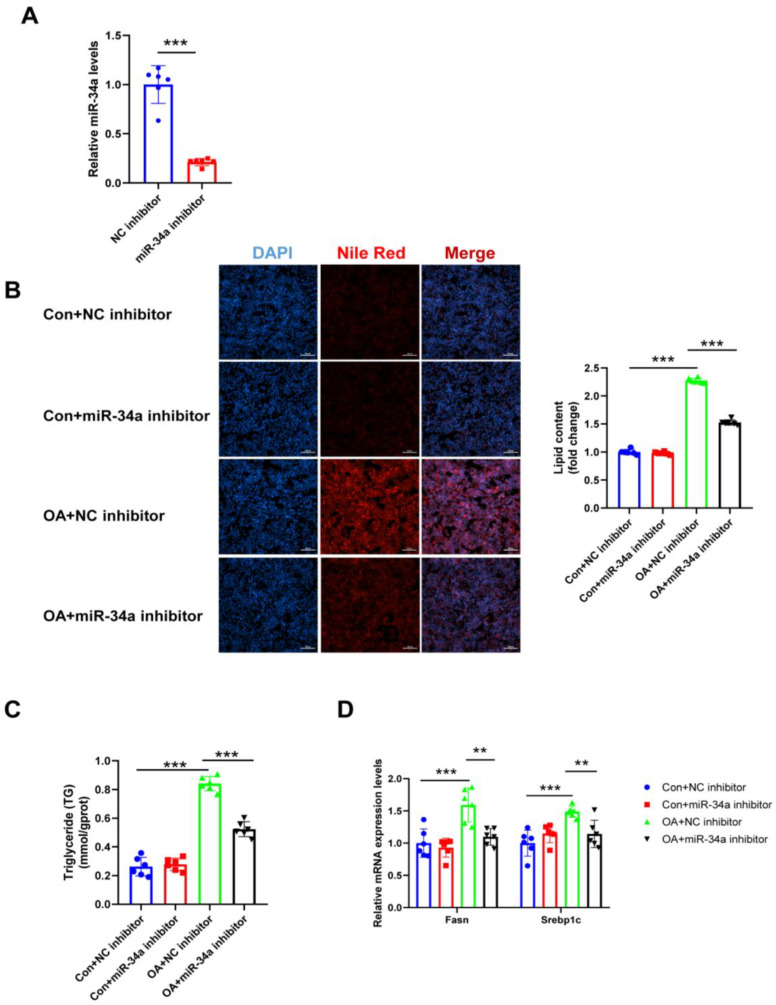
Effects of miR-34a inhibitor on lipogenesis in hepatocytes. (**A**) The expression levels of miR-34a relative to U6 were analyzed by qRT-PCR after transfection with miR-34a inhibitor in HepG2 cells. (**B**) The lipid content of HepG2 cells was determined by Nile Red staining. (**C**) The TG levels of HepG2 cells. (**D**) The mRNA levels of lipogenesis-related genes (Fasn and Srebp1c) relative to GAPDH were analyzed by qRT-PCR. ** *p* < 0.01, *** *p* < 0.001; *n* = 6. OA, oleic acid; TG, triglyceride; Fasn, fatty acid synthase; Srebp1c, sterol regulatory element binding protein-1c. Scale bar = 100 μm.

**Figure 8 nutrients-15-00121-f008:**
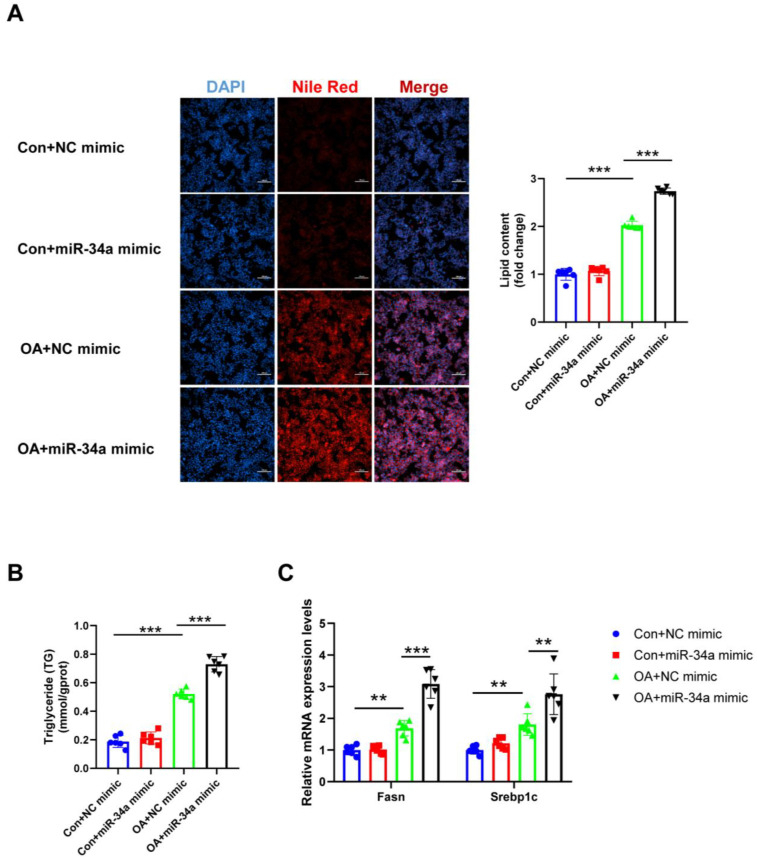
Effects of miR-34a mimicking on lipogenesis in hepatocytes. (**A**) The lipid content of HepG2 cells was determined by Nile Red staining. (**B**) The TG levels of HepG2 cells. (**C**) The mRNA levels of lipogenesis-related genes (Fasn and Srebp1c) relative to GAPDH were analyzed by qRT-PCR. ** *p* < 0.01, *** *p* < 0.001; *n* = 6. OA, oleic acid; TG, triglyceride; Fasn, fatty acid synthase; Srebp1c, sterol regulatory element binding protein-1c. Scale bar = 100 μm.

## Data Availability

The data presented in these studies are available on request from the corresponding author.

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
