# Peer review of "Exercise-Induced ADAR2 Protects against Nonalcoholic Fatty Liver Disease through miR-34a"

_nutrients, 2022, doi:10.3390/nu15010121_

Round 1
Reviewer 1 Report
In this paper, Zhijing Wang and colleagues investigated the role of ADAR in the exercise-induced protection of NAFLD. In particular, the authors showed that exercise can protect the liver by upregulating ADAR2, which participates in liver cell lipogenesis by targeting miR-34a, a crucial regulator of lipid metabolism. Overall, the manuscript is interesting and sheds new light on potential therapeutic strategies in NAFLD. The introduction is appropriate, with sufficient references.
The experiments were designed well, and data interpretation and conclusion are solid
The major problem is represented by the quality of the figures which is very poor. The authors should implement both the magnification and the quality of all figures.
Figure 3. all the abbreviations should be spelled out and reported also in the figure legend (es. OA, FUGW, etc). The very same comment for figure 4.
Material and Methods are adequately described. However, the total number of animals and the number of animals per group are missing. Details regarding the protocol for mice sacrifice and tissue sampling are missing.
Does the fact of using only male animals have a rationale? Would the same experiment done on female mice give comparable results?
Lines 104-108: The authors should add more details and discuss more in depth the aspect regarding the training protocol, with reference for instance to the intensity (percentage of maximal exercise capacity) also referring to other protocols reported in the literature (see for instance: Fernandes et al., Metabolites 2022, 12, 117. https://doi.org/10.3390/metabo12020117; Bartalucci et al. Histol Histopathol. 2012 Jun;27(6):753-69. doi: 10.14670/HH-27.753; Massett et al. Front Physiol. 2021 Dec 2;12:782695. doi: 10.3389/fphys.2021.782695.)
Some typos have been detected (and some of them are also reported, see below). Authors are kindly requested to check the text to ensure that there are no other minor flaws.
Introduction:
Lines 44-50: it is worth adding some other details regarding this aspect; for instance, whether these studies were performed in humans or in experimental animal models.
Line 79: with de novo glioblastoma.
Paragraph 2.3. there are some typos (see for instance: 37oC and CO2). Please correct.
Line 125: “OA (Sigma) for 24h to induce lipogenesis in vitro.”; the abbreviation OA should be spelled out.
Line 146: the cells were fixed; please correct.
Line 148: Cell nuclei were stained; please correct
Author Response
We sincerely thank the reviewer for his valuable feedback that we have used to improve the quality of our manuscript. The reviewer comments are laid out below in italicized font and specific concerns have been numbered. Our response is given in normal font and changes/additions to the manuscript are given in the blue text.
1)The major problem is represented by the quality of the figures which is very poor. The authors should implement both the magnification and the quality of all figures.
Answer 1: We agree with the reviewer’s comments that the quality of figures is not clear. We have replaced the image with new pictures of all figures.
2)Figure 3. all the abbreviations should be spelled out and reported also in the figure legend (es. OA, FUGW, etc). The very same comment for figure 4.
Answer 2: We add the abbreviations in the figure legend of all figures. For example, “HFD”, “L-NAME”,” Fasn” and ”Srebp1c” are added in figure legend 1. “HFD”, “L-NAME” and” ADAR2” are added in figure legend 2. “OE”, “OA”,” ADAR2”, “Fasn” and ”Srebp1c” are added in figure legend 3. “OA”,” ADAR2”, “Fasn” and ”Srebp1c” are added in figure legend 4. “OE”, “OA”, “HFD”, “L-NAME” and” ADAR2” are added in figure legend 5. “OE”, “OA”,” ADAR2”, “Fasn” and” Srebp1c” are added in figure legend 6. “OA”, “Fasn” and ”Srebp1c” are added in figure legend 7 and 8. “FUGW” is a type of vector, which is the full name of it.
3)Material and Methods are adequately described. However, the total number of animals and the number of animals per group are missing. Details regarding the protocol for mice sacrifice and tissue sampling are missing.
Answer 3: We supply the information of total number of animals and the number of animals per group (line 107). And the protocol for mice sacrifice and tissue sampling was added in line 113-116.
4)Does the fact of using only male animals have a rationale? Would the same experiment done on female mice give comparable results?
Answer 4: Thank you for this suggestion. It would have been interesting to explore this aspect. There are three major reasons why we chose the male animals as research subjects.
First, it is estimated that overall prevalence of NAFLD was significantly higher in men than in women (39·7% [36·6-42·8] vs 25·6% [22·3-28·8]; p<0·0001) ( Riazi K et al. Lancet Gastroenterol Hepatol. 2022 Sep;7(9):851-861. doi: 10.1016/S2468-1253(22)00165-0. Epub 2022 Jul 5). Therefore, it is more meaningful for using male animals as research subjects. Second, the physiological and pathological conditions of female animals are more influenced by hormonal profile and body composition (D'Abbondanza M et al. Nutrients. 2020 Sep 9;12(9):2748. doi: 10.3390/nu12092748). Finally, the most research subjects of NAFLD are male animals. (Yang X et al. Hepatology. 2021 Sep;74(3):1319-1338. doi: 10.1002/hep.31863; Li K et al, Metabolism. 2021 Jan;114:154349. doi: 10.1016/j.metabol.2020.154349; Zhi Set al. Int J Biol Sci. 2022 May 1;18(8):3298-3312. doi: 10.7150/ijbs.71431.)
However, sex difference may also influence the effect of exercise. Female mice may show a non-significant response to training. (Massett et al. Front Physiol. 2021 Dec 2;12:782695. doi: 10.3389/fphys.2021.782695; Chen H et al. Signal Transduct Target Ther. 2022 Sep 1;7(1):306. doi: 10.1038/s41392-022-01153-1). So further study is required to determine sex-specific responses upon exercise in NAFLD mice.
5)Lines 104-108: The authors should add more details and discuss more in depth the aspect regarding the training protocol, with reference for instance to the intensity (percentage of maximal exercise capacity) also referring to other protocols reported in the literature (see for instance: Fernandes et al., Metabolites 2022, 12, 117. https://doi.org/10.3390/metabo12020117; Bartalucci et al. Histol Histopathol. 2012 Jun;27(6):753-69. doi: 10.14670/HH-27.753; Massett et al. Front Physiol. 2021 Dec 2;12:782695. doi: 10.3389/fphys.2021.782695.)
Answer 5: Thanks for your great feedback. We explain a little bit more as follows.
In our study, we used moderate intensity aerobic exercise training by treadmill which was similar with our previous research. We have described the exercise training scheme in our methods part detailly (line108-112). In fact, aerobic exercise was an efficient means in alleviating NAFLD in mice, and our research group also reached the same result. Our results demonstrated that exercise was beneficial to reduce liver steatosis and hepatic injury in HF-diet fed mice. The running training reduced bodyweight, attenuated liver steatosis, and restored ALT, AST, TCH and TG serum levels in NAFLD mice (line 340). (Xiao J et al. J Cell Mol Med, 2016, 20(2):204-16. doi: 10.1111/jcmm.12733) ( shown in Figure1).
Besides, it is reported that moderate intensity continuous aerobic exercise does not require adaptive training, which is different from interval high-intensity exercise training. But it needs to ensure uniform movement behavior during exercise (line 341-343). (Bei Y et al. J Sport Health Sci. 2021 Dec;10(6):660-674. doi: 10.1016/j.jshs.2021.08.002).
Finally, the project of moderate-intensity continuous treadmill running varies depending on disease models, functions investigated, and scientific questions posed, such as the daily running duration (30-120 min), running frequency (once or twice per day), and total running duration (weeks to months). It is important to evaluate the exercise efficacy from both its functional protection and its underlying cellular and molecular mechanisms. (Bei Y et al. J Sport Health Sci. 2021 Dec;10(6):660-674. doi: 10.1016/j.jshs.2021.08.002). Therefore, we conducted this exercise training scheme.
6)Some typos have been detected (and some of them are also reported, see below). Authors are kindly requested to check the text to ensure that there are no other minor flaws.
Answer 6: We feel sorry for our carelessness. In our resubmitted manuscript, the typo is revised. Thanks for your correction.
Reviewer 2 Report
The study is interesting and contains novel insights into the mechanisms of beneficial effects of exercise in NAFLD.
The language, however, is exceptionally poor and needs to be reviewed by a native English-speaking professional before being reviewed on behalf of its scientific content. The reviewer cannot dedicate his time on syntax, spellings and other language problems in a manuscript in which almost every phrase requires correction and some sentences are difficult to understand. The reviewer, hence, stopped the reviewing process after having gone through the abstract.
When abbreviations are used, they need to be written in full when used for the first time.
Line 12:
approach to NAFLD
Line 13-15:
In this study, mice on a normal diet or high-fat diet (HFD) combined with Nω-Nitro-L-arginine methyl ester hydrochloride (L-NAME) were either kept sedentary or subjected to a 12-week exercise running scheme …
Line 15-16:
We found that exercise reduced liver steatosis in mice with diet induced NAFLD.
Line 16:
Hepatic adenosine deaminase
Line 17:
RNA 2 (ADAR2)
Line 17:
was downregulated in NAFLD and was upregulated in the liver after 12-week exercise.
Line 18-19:
In HepG2 cells treated with OA (??), overexpression of ADAR2 inhibited and suppression promoted lipogenesis.
Line 19-20:
… ADAR2 could down regulate mature miR-34a in hepatocytes.
Line 20-21:
“Functional reverse experiments further proved this by performing through miR-34a mimic and ADAR2 overexpression” requires more specific rephrasing to make it better understood.
Line 21-22:
Moreover, miR-34a inhibition and mimicry could also affect lipogenesis in hepatocytes.
Author Response
According to the associate reviewer’ comments, we have made extensive modifications to our manuscript. Thank you again for your positive comments and valuable suggestions to improve the quality of our manuscript. The specific modifications were listed below in points.
- We apologize for the poor language of our manuscript. We worked on the manuscript for a long time. The repeated addition and removal of sentences obviously led to poor readability. We have now worked on both language and readability and have also polished the language by the editing service to improve readability of the manuscript. The English editing ID is “English-54742”.
- Besides, we have corrected the words and sentences of abstract according to the comments of the reviewer.
- We have implemented the useful observations of reviewer about the abbreviations. All abbreviations in the paper are written in full when used for the first time.
Round 2
Reviewer 1 Report
Although the Authors have answered most of the concerns raised by the reviewer, there are still some problems with the figures. In fact, the Authors stated that they have “replaced the image with new pictures of all figures”. However, it seems only that they have increased the size of the pictures, but the quality of the figures is still very poor. See for instance Fig. 1C and 1D, Fig. 3B, Fig. 4B, Fig. 6B, Fog. 7B, Fig. 8B.
Moreover, I did not find the answer to the present question and the relative comment within the revised manuscript: “Lines 44-50: it is worth adding some other details regarding this aspect; for instance, whether these studies were performed in humans or in experimental animal models.”
Author Response
We feel great thanks for your professional review work on our article. As you are concerned, there are still several problems that need to be addressed. According to your nice suggestions, we have made extensive corrections to our previous draft, the detailed corrections are listed below.
1)We agree with the comment that the in figures are not clear, especially of Fig. 1C and 1D, Fig. 3B, Fig. 4B, Fig. 6B, Fig. 7B, Fig. 8B. We have replaced the images with new figures in high quality.
2)We think this is an excellent suggestion. We have added some details in the content of line 48 and line 50, which are marked in blue in the revised paper. We could see that the studies were performed in NAFLD patients.
Reviewer 2 Report
The study is interesting and contains novel insights into the mechanisms of beneficial effects of exercise in NAFLD.
The methods are appropriate for examining the hypothesis of the authors.
The language, however, still has to be improved.
Line 17-18:
and were upregulated in the liver after 12-week exercise
Line 20:
“ … oleic acid (OA), respectively“
Line 20-22:
The sentence “Functional reverse experiments further proved miR-34a mimicry eliminated the suppression of ADAR2 overexpression in lipogenesis in vitro.” still does not make sense. It requires rephrasing. Do the authors want to say: “Functional reverse experiments further proved that miR-34a mimicry eliminated the suppression of ADAR2 overexpression in lipogenesis in vitro.”?
Line 29:
“ …has economic impact“
Line 41- 44:
Better rephrase: “Exercise can alleviate NAFLD by regulating lipid metabolism, lipophagy, insulin resistance, and inflammatory response …”
Line 53:
“… RNA (ADAR) …”
Line 58:
“… RNA editing enables generation of thousands of different RNAs …”
Line 73:
“MiRNAs , including miR-122, …”
Line 83:
“MiR- …”
Line 94:
“… strategy for managing NAFLD.”
Line 96:
“2.1 Animals and treatments”
Line 100-101:
A reference for the rational of using L-NAME should be given, e.g., “ … and Nω-nitro-L-arginine me-100 thyl ester, hydrochloride (L-NAME) for 12 weeks, a NOS inhibitor, which was shown to exacerbates liver injury in an obese rodent model of NAFLD (Sheldon et al., Am J Physiol Gastrointest Liver Physiol 308: G540–G549, 2015).”
Line 123:
“Frozen 4μm liver sections …”
Line 130:
“ … was used as a …”
Line 132:
37°C
Line 148:
“. MiR-34a inhibitor …”
Line 160:
The principle of the assay for measuring triglycerides should be given.
Line 191:
“in mice on HFD + L-NAME compared to the controls”
Fig. 1A:
Instead of “Run” and “HFD + L-NAME” better use the wording “intervention”, since this term refers to all 4 groups.
Fig. 1E:
The ordinate should be “Gene expression”. The wording relative to GAPDH should be given in the legend in
Line 205:
“… genes related to liver lipogenesis (Fasn and Srebp1c) relative to GAPDH in the livers of mice …”
Line 198:
“… levels of these genes were …”
Line 210:
“… in the group with diet-induced NAFLD”
Line 213-214:
“decreased in NAFLD and could …”
Line 216:
“that ADAR2 expression was inhibited in NAFLD”
Line 216-217:
“could be induced by exercise, which was associated with hepatic lipid accumulation suggesting a role of ADAR2 in lipid metabolism.
Line 219:
“ … in the group with diet-induced NAFLD.”
Line 227:
“ … analyses were performed”
Line 238:
“ … provide evidence …”
Line 271:
“in the livers of mice with NAFLD“
Line 276:
“We further examined whether …”
Line 280:
“ … miR-34a mimicking significantly …”
Line 283:
“… while miR-34a mimicking could reverse …”
Line 297:
“First, miR-34a was downregulated …”
Line 298-299:
“As a result, miR-34a inhibitor decreased the lipid content (Fig. 7B-C), whereas miR-34a mimicking raised the lipid content …”
Legends:
Figure 1 not Figure1
Figure 2
etc.
Line 313:
“Effects of miR-34a mimicking on lipogenesis …”
Line 328:
“aerobic exercise”
Line 337:
“cannot”
Line 341-342:
“has been confirmed in animal models by several studies, and our previous study essentially showed the same result …”
Line 345:
The syntax is not correct. It is not clear what the authors want to express linking reducing the Fasn and Srebp1c with reduction in hepatic lipogenesis. The reference 40 only supports reduction of lipogenesis.
“… was able to reduce hepatic lipogenesis …”
Line 347:
compound- and diet-induced NAFLD
Line 376:
“needs elucidation by further research”
Line 379:
“First, ADARs edit pri-miRNA or pre-miRNA …”
Line 391:
“To define the relationship of ...”
Line 402:
nonalcoholic steatohepatitis
Line 405-406:
hepatocyte nuclear factor 4, alpha (HNF4α), sirtuin1(Sirt1), and cyclin-dependent kinase 6 (CDK6)
Line 214-216:
The phrase requires reconstruction, e.g., “ further study is required to carry out in vivo research in mice is required to evaluate the effect of ADAR2 as a therapeutic method for NAFLD.”
Line 414-415:
Phrasing is inconsistent. Suggestion by the reviewer:
“is also unclearrequires further clarification. This could advance our understanding …”
